



# A fast sample shuttle to couple high and low magnetic fields. Applications to high-resolution relaxometry

Jorge A. Villanueva-Garibay,[1,*] Andreas Tilch,[1] Ana Paula Aguilar Alva,[2] Guillaume Bouvignies,[2] Frank Engelke,[1] Fabien Ferrage,[2,*] Agnes Glémot,[3] Ulric B. le Paige,[2] Giulia Licciardi,[4,5,6] Claudio Luchinat,[4,5,6,*] Giacomo Parigi,[4,5,6] Philippe Pelupessy,[2] Enrico Ravera,[4,5,6] Alessandro Ruda,[2] Lucas Siemons,[2] Olof Stenström,[2] Jean-Max Tyburn[7]

[1]Bruker BioSpin GmbH, Rudolf-Plank-Str. 23, 76275 Ettlingen, Germany
[2]Laboratoire des Biomolécules, LBM, Département de chimie, École normale supérieure, PSL University, Sorbonne Université, CNRS, 75005 Paris, France
[3]Bruker BioSpin AG, Industriestrasse 26, 8117 Fällanden, Switzerland
[4]Department of Chemistry "Ugo Schiff", University of Florence, via della Lastruccia 3, Sesto Fiorentino, 50019 Italy
[5]Magnetic Resonance Center (CERM), University of Florence, via Sacconi 6, Sesto Fiorentino, 50019 Italy
[6]Consorzio Interuniversitario Risonanze Magnetiche MetalloProteine (CIRMMP), via Sacconi 6, Sesto Fiorentino, 50019 Italy
[7]Bruker BioSpin ; 34 rue de l'Industrie BP 10002, 67166 Wissembourg Cedex, France

*Correspondence to*: Jorge A. Villanueva-Garibay (Jorge.Garibay@bruker.com); Fabien Ferrage (Fabien.Ferrage@ens.psl.eu); Claudio Luchinat (luchinat@cerm.unifi.it)

**Abstract.** Combining high-field high-resolution NMR with an evolution of spin systems at low magnetic field offers many opportunities for the investigation of molecular motions, hyperpolarization and the exploration of field-dependent spin dynamics. Fast and reproducible transfer between high and low fields is required to minimize polarization losses due to longitudinal relaxation. Here, we introduce a new design of a sample shuttle that achieves remarkably high speeds, $v_{max} \sim 27$ m.s$^{-1}$. This hybrid pneumatic/mechanical apparatus is compatible with conventional probes at the high-field center. We show applications to water relaxometry in solutions of paramagnetic ions, high-resolution proton relaxometry of a small molecule and sample shuttling of a solution of a 42 kDa protein. Importantly, this fast sample shuttle system is narrow, with a diameter $d = 6$ mm for the sample shuttle container based on a standard 5 mm outer diameter glass tube, which should allow near access to the sample for magnetic manipulation at low field.

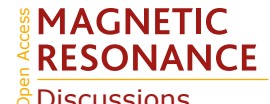

# 1 Introduction

High-resolution nuclear magnetic resonance (NMR) is preferably performed at high magnetic fields, with applications to chemistry, structural biology, material sciences or metabolomics. On the other hand, many types of NMR experiments require or benefit from access to low magnetic fields. Relaxometry probes molecular motions by quantifying relaxation over a broad range of magnetic fields, typically down to 100 µT (Kimmich and Anoardo, 2004). Many hyperpolarization methods can also benefit from low magnetic fields, such as CIDNP (Grosse et al., 1999; Li et al., 2023), parahydrogen-induced polarization

(Pravdivtsev et al., 2015) or Overhauser DNP (Ravera et al., 2016; Reese et al., 2009), or from cycling between high and low magnetic fields (Ivanov et al., 2014). The relaxation rates of long-lived states (Carravetta et al., 2004) long-lived coherences (Pileio et al., 2009; Sarkar et al., 2011), and multiple-quantum coherences under chemical exchange (Cousin et al., 2016b) are diminished at low magnetic fields that reduce the chemical-shift interaction. Zero- and ultra-low field methods require even lower magnetic fields, in the nT range, to make the nuclear Zeeman interactions smaller than scalar couplings (Barskiy et al.,

2024). The analytical power of high-resolution high-field NMR can be combined with all these approaches if an apparatus can transfer the sample between the magnetic center of a high-field spectrometer and a position of low field, typically over a distance of the order of a few cm up to *ca.* 1 m.

Physically moving the sample between high and low-field positions takes time, usually tens to hundreds of milliseconds. Such

transfer times are particularly limiting for applications to molecular systems with fast longitudinal relaxation, such as nuclear spins in paramagnetic complexes, which relax fast at all magnetic fields, or in macromolecules, where fast longitudinal relaxation is encountered at low magnetic fields. Coupling high and low magnetic fields to such systems requires to move the sample as fast as possible. It follows that sample shuttles are preferable to probe-shuttle systems (Grosse et al., 1999; Victor et al., 2004), as the inertia of the probe makes the displacement of the sample necessarily slow. Sample shuttles are moved

either with a mechanical system, where the action of a motor is transferred to the sample by a rope (Miéville et al., 2011; Pileio et al., 2010), a belt (Chou et al., 2012; Redfield, 2012), ~~or~~ a rack (Kiryutin et al., 2016), or a pneumatic system (Charlier et al., 2013; Kerwood and Bolton, 1987; Redfield, 2003; Reese et al., 2008). Pneumatic systems take advantage of moving a sample shuttle of limited size and inertia and rely on easily manageable pressures. Yet, the high-field landing must be accommodated, possibly by a specially designed probe, which limits the range of applications and, most importantly, the

sensitivity (Charlier et al., 2013). In addition, the hard landings of pneumatic sample shuttles can be detrimental to both the hardware and fragile molecular systems. Most motor-operated systems offer the ability to operate from the top, which allows the use of conventional probes and can reach state-of-the-art sensitivity in mechanical systems (Chou et al., 2016). In addition, the control of the trajectory at all times prevents the occurrence of extreme forces at the end of the sample shuttle path. Yet, the use of a belt or rack in some mechanical systems makes the sample shuttle bulky and prevents close access to the sample

at low field. This can be a limitation for applications such as two-field NMR (Cousin et al., 2016b, a; Kadeřávek et al., 2019; Robertson et al., 2023) where the sample shuttle has to move through radiofrequency and gradient coils. The last version of



the Redfield shuttle (Redfield, 2012) featured a sample shuttle with small dimensions along the transverse axes with a long stick that coupled the sample to a belt and pulley system sitting on top of the high-field magnet. Yet, such a design requires very high ceilings in the laboratory, and can be a source of large vibrations throughout the magnet.


Our objective is to develop a new type of sample shuttle system that combines high speed, full control of the trajectory and small transverse dimensions. Here, we introduce a hybrid pneumatic/mechanical sample shuttle system, able to reach high speed ($v_{max} \sim 27$ m.s$^{-1}$) with full control of the trajectory (Aders et al., 2022). The principal components of this system are shown in Figure 1. The diameter of the sample container is $d = 6$ mm (5 mm for the NMR tube alone), allowing close access

to the sample at low fields for a range of applications. We present the design of this fast sample shuttle and achievable specifications for the motion of the sample shuttle. We illustrate the use of the fast sample shuttle in a broad selection of samples: we compare water proton relaxometry in solutions of paramagnetic ions with fast field-cycling relaxometry. We measure high-resolution relaxometry of a small molecule that transiently binds to a macromolecule, and show the spectral quality and sensitivity of a two-dimensional correlation in a methyl-labeled protein.

## 75   2 Hardware

The simplest way to move a sample tube out of a vertical high-field NMR magnet is to use a motor and a rope to pull the sample up. Several versions of such a system, from rudimentary to sophisticated designs, have been built in several laboratories (Hall et al., 2020; Miéville et al., 2011; Pileio et al., 2010). As long as the rope is tense, the position of the sample is controlled at all times. The Achilles heel of such a system is the weak acceleration offered by gravity to move the sample back down.

Here, we solve this limitation by using constant pressure applied from the top of the system. We show a scheme of the principle of this hybrid pneumatic/mechanical system in Figure 1, and list the characteristic parameters of this fast sample shuttle in Table 1.

### 2.1 Overall design

The fast sample shuttle system includes the following parts, which we will describe below:

• A drive unit that includes two 800 W servomotors; a winch wheel and an inertia compensation disc, 3.5 bar pressurized gas supply.

• The drive unit is controlled with a set of sensors and communicates with the spectrometer with real time trigger signals and with the workstation PC carried by RS232 data exchange.

• A glass shuttle guiding tube, with a dampening system at the high-field end.

• A sample transfer station with an NMR sample container access window.

• A sample container, which consists of a 9-inches NMR tube, with two guiding sleeves, and an end cap attached to a rope.



**Table 1: Characteristic parameters of the fast sample shuttle systems as currently installed.**

| Parameter | Values |
|---|---|
| Maximum shuttle distance | 1020 mm (at 700 MHz), 800 mm (at 600 MHz) |
| Minimum shuttle distance | 50 mm (> 99.97% of max. field) |
| Gas pressure | 3.5 bar |
| Gas flow rate range | 5 to 20 L/min |
| Outer diameter of NMR sample tube | 5 mm |
| Maximum shuttle acceleration | 1112.6 m/s$^2$ (113.5 × g) |
| Maximum shuttle speed | 26.7 m/s |
| Shuttle travel time (1020 mm) | 68.5 ms (see Fig. 6) |
| Shuttle travel time (800 mm) | 61 ms |
| Vibrational artifacts in $^1$H NMR spectrum* | < 2% peak to peak |

* Evaluated on the residual water proton resonance in D$_2$O

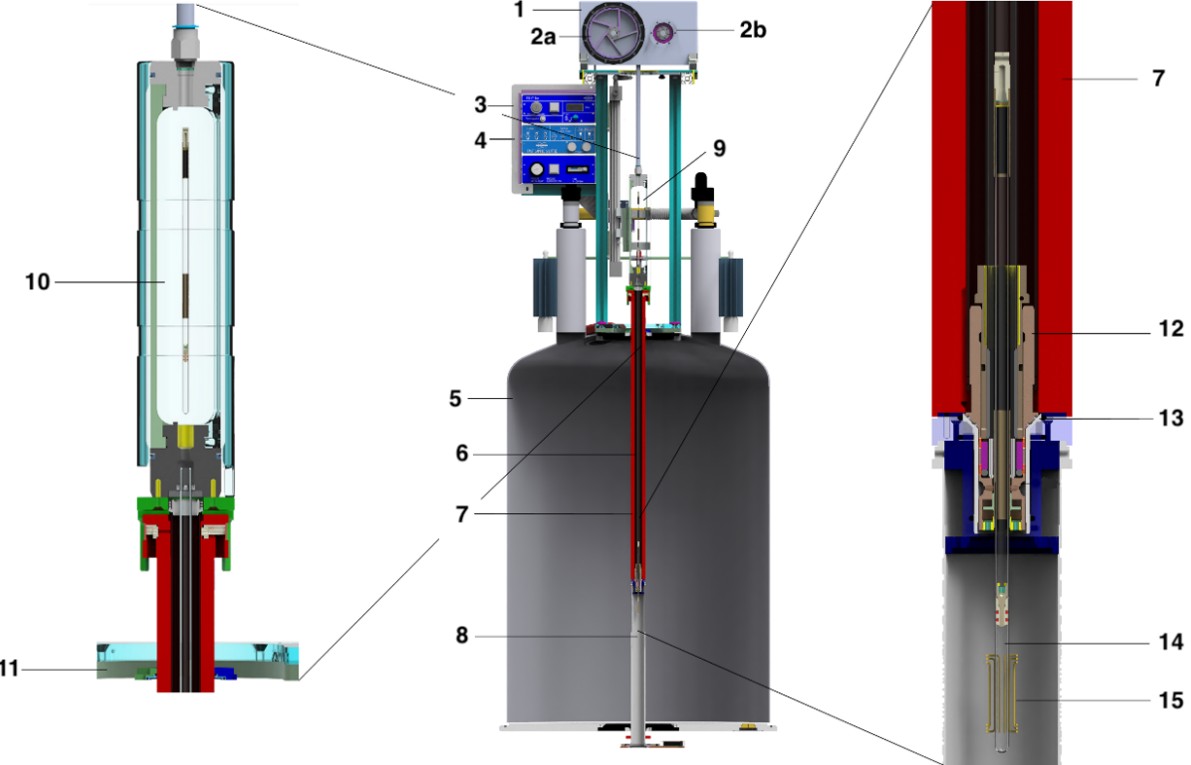

**Figure 1. Schematic representation of the fast sample shuttle (center). Details of the sample transfer station and of the end-stopper assembly at the high-field position to the left and right of the center scheme, respectively. The components presented are: 1 – drive**



**unit, 2a – winch wheel, 2b – reversely rotating disc, 3 – pneumatic unit, 4 – electronic unit, 5 – NMR magnet, 6 – shuttle guiding**
**tube, 7 – shim upper stack, 8 – shim system and NMR probe, 9 – sample transfer station, 10 – window for sample transfer, 11 –**
**mounting flange, 12 – end-stopper, 13 – NMR sample turbine, 14 – NMR sample container, 15 – NMR rf coil**

The fast sample shuttle (FSS) shown in Fig. 1, is an accessory that can be fitted in any Bruker high-resolution NMR magnet (5). It uses the already available shim upper stack (7) inner bore to accommodate the shuttle guiding tube (6). This guiding tube interfaces at the top with the sample transfer station (9) and with the NMR sample turbine (13) at the bottom. The NMR
sample turbine hosts the end-stopper assembly (12). Inside the guiding tube the NMR sample container (14) glides at very high speeds up and down between the limits imposed by the NMR magnet geometry. Those limits are on one side the NMR rf coil (15), inside at the upper part of the NMR probe (8). At this point, the NMR sample container places the sample active volume at the same position as any standard NMR 7-inches glass tube. Therefore, the amount of sample needed for high-resolution relaxometry (HRR) experiments is similar to that used for static NMR experiments in conventional 5 mm tubes.
The other limit depends on the length of the magnet, measured from its center to almost its upper closing lid (11). For instance, for an ultra-shielded 600 MHz magnet this distance is $d_{max600} = 800$ mm with a field $B_{low} = 36.6$ mT. Similarly, for an ultrashield 700 MHz magnet it is $d_{max700} = 1020$ mm, with a field at this position $B_{low} = 46.6$ mT. On the other end, the minimum travel distance was set to, but is not limited to, $d_{min} = 50$ mm. This position is at the border of the homogeneous magnetic field plateau with a magnetic field at 99.97% of the maximum magnetic field at the magnetic center. Shorter travel distances would give
access to the same magnetic field as that at the center of the magnet (static NMR condition). It is important to note that the FSS is compatible with any high-resolution NMR probe-head.

The sample container is introduced inside the magnet through the sample transfer station (9,10). This process is done by mechanically moving upwards a cylindrical PMMA (poly methyl methacrylate) window for the sample transfer (10) that is
attached to a sliding mechanism. At the bottom of the sample transfer station there is an access to the guiding tube. The sliding mechanism for the sample transfer window is set to a rigid supporting structure that in turn is anchored to the mounting flange (11) at the top of the magnet. The drive unit (1), which includes two motors, is located at the top of the supporting structure. The main motor has a winch wheel (2a) and it is in charge to move the sample container. A secondary motor, that synchronously counter moves with respect to the main motor, contains a reversely rotating disc with the same inertia as the winch wheel (2b).
The function of the secondary motor is to minimize mechanical vibrations caused by the movement of the main motor.

To ensure fast deceleration on the way to low fields or efficient acceleration on the way back to the high-field center, gravity is not sufficient. A strong downward force is obtained with air pressure. The pneumatic unit (3) provides the regulated gas pressure and flow. The FSS working pressure is 3.5 bar with a gas flow below 20 liters-per-minute. The electronic unit (4)
takes care of all FSS safety sensors and provides the communication interface between the motors and the NMR console. In





addition, it powers the extension module of each motor as well as the pneumatic unit. Figure 2 is a photograph of the the external components of the FSS system built at CERM.

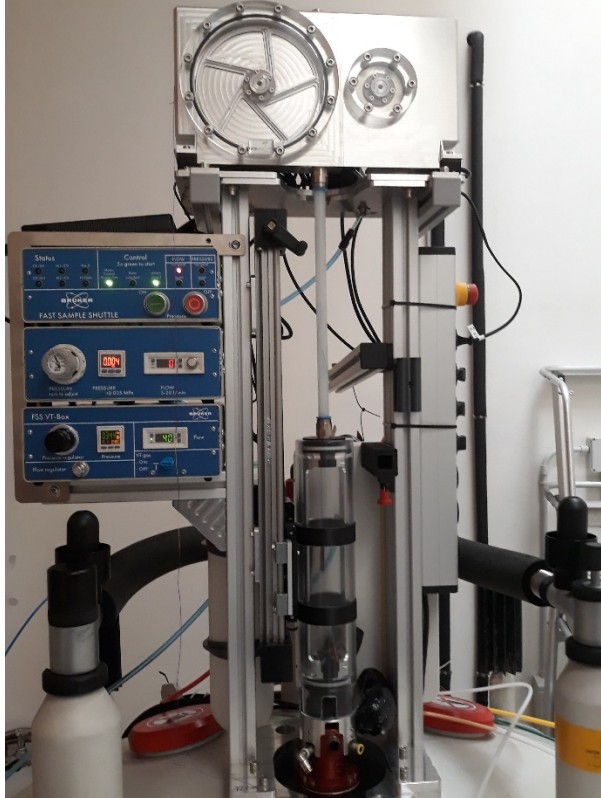

**Figure 2. Photograph of the FSS installed on the 700 MHz NMR spectrometer at the CERM in Florence, Italy.**

The top of the photograph (Figure 2) shows the two-motors configuration used to compensate mechanical vibration artefacts. Since the vibrations caused by a single motor/winch wheel are coherent, the second motor/disc in reverse operation is used to cancel out the vibrations of the first. Some vibrations remain after the sample container has landed at the end-stopper, requiring a short stabilization delay. After a stabilization delay $\tau_{st} = 150$ ms, vibration artefacts in proton spectra are less than 2% of the peak intensity (see below). Figure 3 shows the diagram of the upper part of sample container (14) under pressure (p). In order

to ensure the full control of the trajectory, the shuttle rope (17) that is connected to the end cap (16) and, in turn, to the sample container (14) must be always under tension during operation. This requires that the force exerted by the pressurized gas be higher than the product of the mass of the sample container and the maximum acceleration. With a top surface area $s_{top} = 28$ mm$^2$ and a mass $m_{container} = 4.7$ g, an acceleration a $= 100 \times g$ is compensated by a pressure P $\sim 1.5$ bar. The gas pressure chosen for the current design is 3.5 bar. The sample container position is indirectly monitored and precisely controlled by the angular

variation of the servo motor (800W servo motor, JVL MAC800, https://www.jvl.dk/).



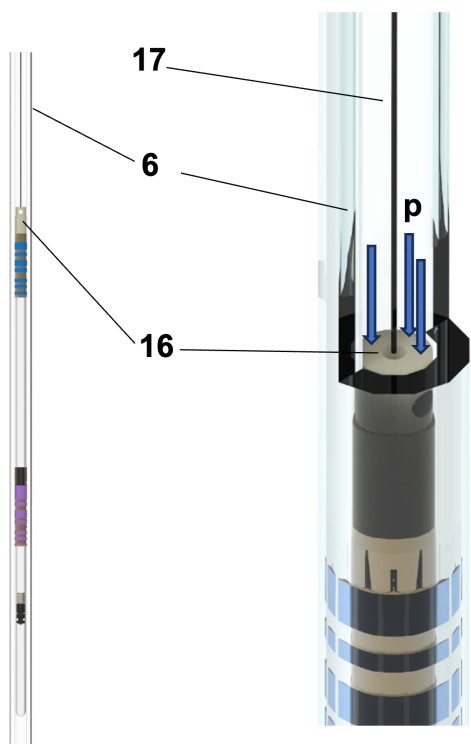

**Figure 3. Diagram of sample container in the guidance tube with gas pressure (p) applied. 6 – shuttle guiding tube (borosilicate 3.3 glass), 16 – sample container end cap, 17 – shuttle rope (polyethylene).**

Figure 4 depicts in detail the components of the sample container (14). A 9-inches long borosilicate 3.3 glass with a 5 mm outer diameter is the base of the sample container (14a). This thin-wall NMR tube has two Vespel$^{TM}$ sleeves, one at the upper end (14b) and another one (14c) positioned to balance the assembly. The sample volume (14f) is about 650 µl and it is confined by a two-component seal plug (14d and 14e). The plug allows to remove trapped air and keeps the sample positioned at the bottom of the tube. The end cap (16) has a small orifice that allows to keep the sample plug under the working pressure. The

inner part of the upper sleeve (14b) of the sample container is threaded so that it can be screwed to the base of the end cap. The rope (17) is connected to the end cap by a knot that tightens upon tension.



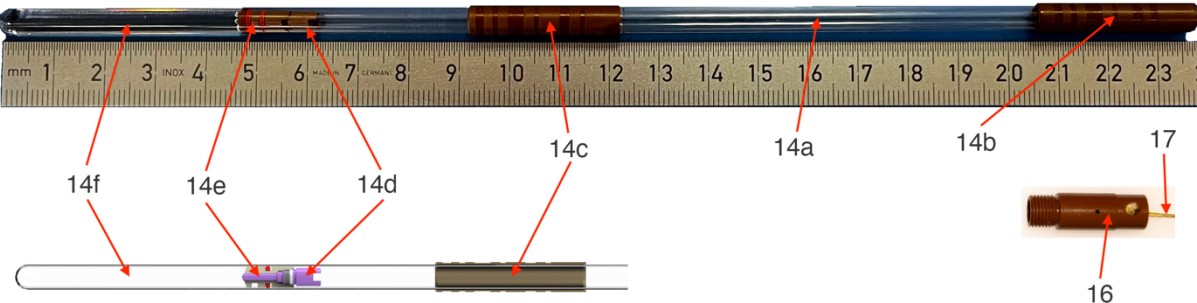

**Figure 4. Sample container components. 14a – high precision thin wall NMR tube with diameter d = 5 mm and length l = 9 inches,**
**14b – upper sleeve with thread, 14c – lower sleeve, 14d and 14e – plug elements to limit the sample volume 14f, 16 – end cap with**
**shuttle rope (17) attached.**

The different communication paths that are needed for the function of the fast sample shuttle are shown in Figure 5. A control program (AU program) run from the workstation that controls the spectrometer centralizes all communication channels. This AU program tells the motors when and how to move and provides them with relevant information about the position and acceleration as well as several functionalities to simplify the handling of the FSS. This channel of communication uses a bidirectional RS232 serial-port protocol, that reads and writes motor registers. In addition, the AU control program calls the NMR pulse sequence and activates real time signals (TTL) from the NMR console. Those signals are sent towards the electronic unit (4 in Figure 1) and transferred to the motors if the operational conditions are met. The operational conditions are monitored by the electronic unit. This unit checks the status of the motors and sample transfer station (closed), the pressure and flow rate values and whether the shuttle rope is unclamped. When the sample container arrives at the target low-field position, the motors send a trigger signal back to the NMR console, which is then transferred to the workstation via ethernet. At the end of the predefined relaxation delay time at the low-field position, the sample container is sent back to the high-field magnetic center and the communication process is repeated for the following transient.

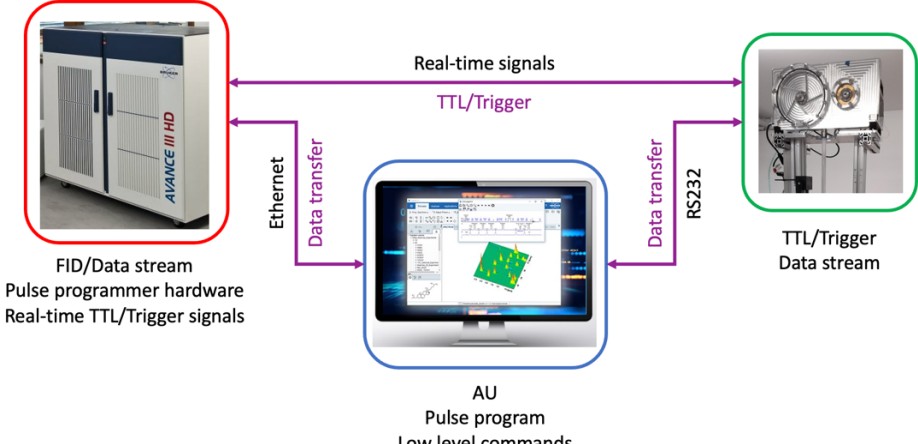

**Figure 5. Data communication scheme among the workstation PC, NMR console and driving motors.**





## 2.2 Trajectory of the sample container

Full control of the trajectory of the sample container (position as a function of time) is essential for reproducibility and to ensure the acceleration is kept within boundaries tolerable by the sample and spectrometer. The "soft landing" of the sample container at the high-field magnetic center is critical to minimize vibrations and to avoid additional shock loads to the system.

As an example, Figures 6 and 7 show experimental data of the FSS trajectory as obtained on the ultrashield 700 MHz NMR system at CERM in Florence, for a full shuttle cycle. The motion starts at time zero at the high-field position (sample in the NMR probe), then moves up 1020 mm to the furthest low-field position within a time $\tau_{UP}$ = 68 ms (Shuttle UP). The sample resides at the low-field position, for a set delay, here $\tau_{LF}$ = 10 ms. Note that the minimal value of the residence time at low field is $\tau_{LFmin}$ = 3 ms. Subsequently, the sample is shuttled down back to the high-field position again during $\tau_{DOWN} = \tau_{UP}$.

Similar profiles were obtained for the ultrashield 600 MHz NMR systems at ENS and at Bruker. In this latter case, the travel distance is shorter (800 mm), leading to shorter transfer durations $\tau_{DOWN,600} = \tau_{UP,600}$ = 61 ms. The increment in transfer duration is only ~8 ms for an increase of the distance of 220 mm as the extra distance is covered at highest speed under zero acceleration. The acceleration is constant for a given fixed travel time between velocity zero and maximum velocity (see Figures 6 and 7). Of course, the sign of acceleration changes depending on whether the shuttle motion is going up or down.


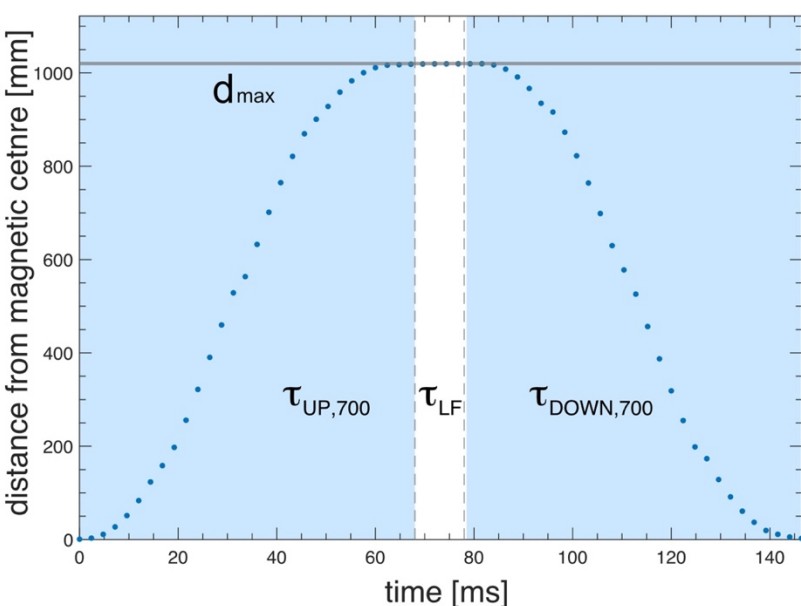

**Figure 6. Measured shuttle trajectory (travel distance vs. time) for the FSS installed at the 700 MHz NMR spectrometer at CERM in Florence.**





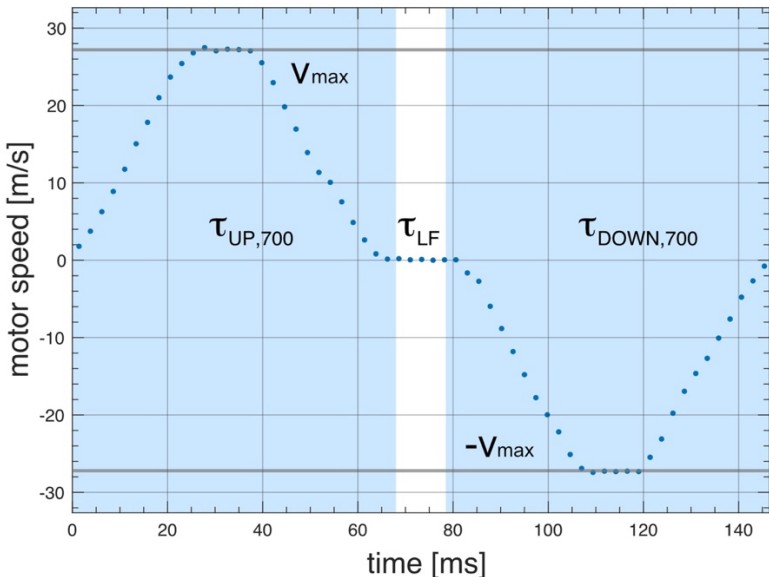

**Figure 7. Measured shuttle trajectory (motor speed vs. time) for the FSS installed at the 700 MHz NMR spectrometer at CERM in Florence. The relaxation time at the low field position is $\tau_{LF}$.**

The trajectories of the sample container follow the motion with constant acceleration up to the middle of the trajectory or a predefined maximum speed $v_{max}$, as experimentally demonstrated in Figures 6 and 7. Therefore, as long as the maximum speed is reached only for a small fraction of the path, for a given travel time, the motor acceleration is proportional to the travel distance. The minimum travel time was determined to be $\tau_{UP,700} = 68$ ms for $d_{max} = 1020$ mm and $\tau_{UP,600} = 61$ ms for $d_{max} = 800$ mm. By keeping constant the travel time, the motor acceleration was measured for all other distances plotted in Figure 8. The experimental data was linearly fitted along with a 95% of prediction interval of confidence (solid and dashed lines, respectively). Similar results were obtained for the NMR system at ENS in Paris.





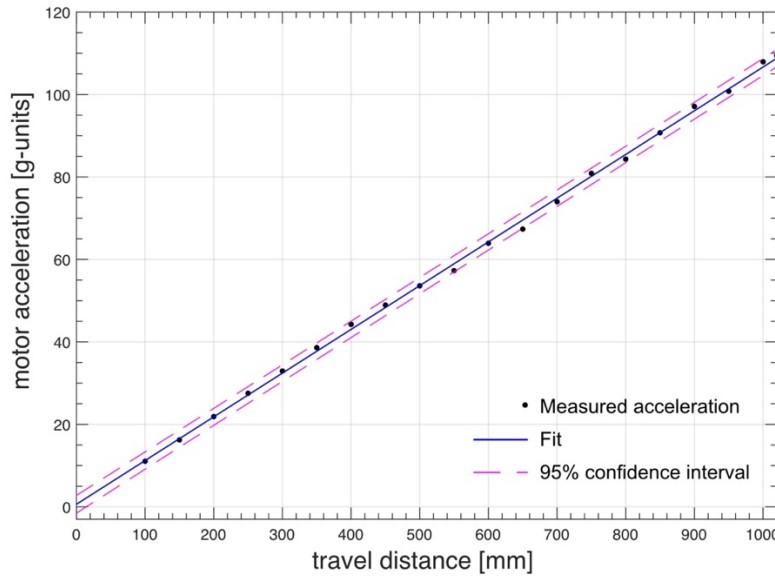


**Figure 8. Measured motor acceleration vs. travel distance at constant travel time ($\tau_{UP,700}$ = 68 ms) for the FSS installed at the 700 MHz NMR spectrometer at CERM in Florence.**

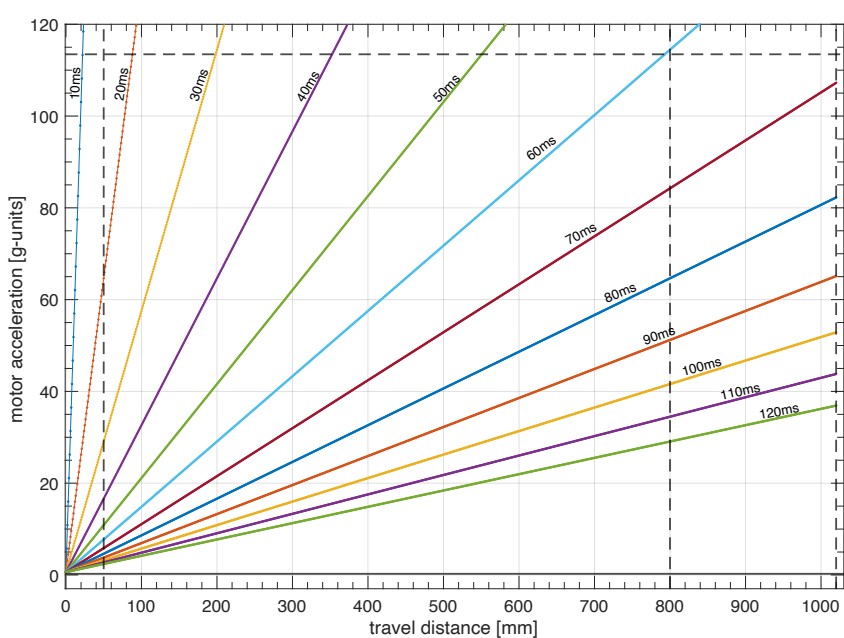

**Figure 9. Achievable shuttle travel times as function of travel distance and motor acceleration.**



Polarization losses due to relaxation during transfers between high and low fields should be minimized. Could the delays $\tau_{UP}$ and $\tau_{DOWN}$ be significantly reduced? The transfer delays $\tau_{UP}$ = 61 ms for a distance $d_{max}$ = 800 mm and $\tau_{UP}$ = 68 ms for a distance $d_{max}$ = 1020 mm require to reach the maximum acceleration achievable $a_{max}$ = 113 × g (Figure 9). The acceleration increases quadratically with decreasing transfer delays. Reaching slightly shorter transfer times, for instance $\tau_{UP}$ = 50 ms for a

distance $d_{max}$ = 800 mm would require an acceleration $a_{max}$ = 150 × g. Much shorter duration of transfer of the sample shuttle would require accelerations impossible to achieve with the current design.

### 2.3 Effects of vibrations on proton spectra

The forces generated by the motors (2a and 2b in Fig. 1) lead to vibrations that propagate throughout the Dewar and NMR probe, despite the compensation provided through their reverse motion, and minimal disturbances of the sample container

motion. As a consequence, after landing of the sample shuttle container in the NMR probe, it is necessary to insert a waiting period (stabilization time $\tau_{st}$) to allow the decay of residual vibrations. These residual vibrations lead to artifacts in $^1$H NMR spectra in the form of vibrational sidebands, with frequency distances ranging from 7 to 60 Hz around the main NMR line (Figure 10). On the FSS system installed on the 700 MHz NMR spectrometer available at CERM, the total amplitudes of these sidebands are at most 2.2 % (peak to peak) of the peak height after a stabilization delay as short as $\tau_{st}$ = 50 ms after shuttle

motion from the maximum distance $d_{max}$ = 1020 mm. Similar vibration profiles and vibration amplitudes were observed on the FSS system installed on the 600 MHz NMR spectrometer located at Bruker in Wissembourg, France. However, those vibration amplitudes were significantly higher on the FSS system installed on the 600 MHz NMR spectrometer at ENS in Paris, in spite of manufacturing within the tight tolerances of all the system components as the one in Wissembourg.



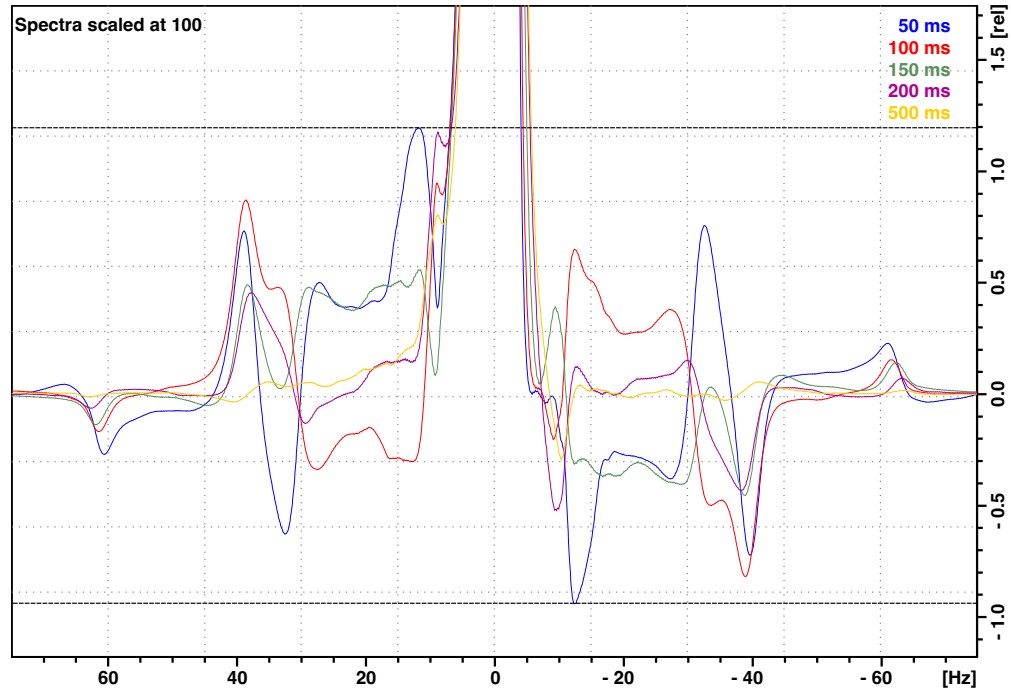


**Figure 10. $^1$H NMR spectra of 99.8% D$_2$O (residual water peak 0.2% H$_2$O) with the vertical axis in percent of the peak maximum showing vibrational sideband artifacts at the base of the NMR peak. The various spectra indicate the vibrational artifacts after stabilization times (time interval between shuttle landing and 90° pulse) $\tau_{st}$ = 50 (blue), 100 (red), 150 (green), 200 (burgundy), and 500 ms (orange). Data were taken on the 700 MHz spectrometer installed at CERM in Florence with the maximum 1020 mm travel distance.**

A stabilization delay $\tau_{st}$ is inserted in pulse sequences to reduce vibrations artefacts in spectra. This stabilization delay leads to polarization losses due to longitudinal relaxation. How long should the stabilization delay be? Vibration artifacts decay with increasing stabilization delay $\tau_{st}$ (Figure 11.a-b). With a maximum travel distance of 1020 mm, vibration artefacts are small, with a peak to peak of about 1% of the peak amplitude after 150 ms and mostly fade out after 500 ms. As the acceleration is proportional to the travel distance, the forces exerted by the motors on the structure increase with increasing travel distance, leading to increased level of vibrations (Figure 11.c-d). The stabilization time was set to 150 ms and then the travel distance was varied from 100 to 1020 mm.





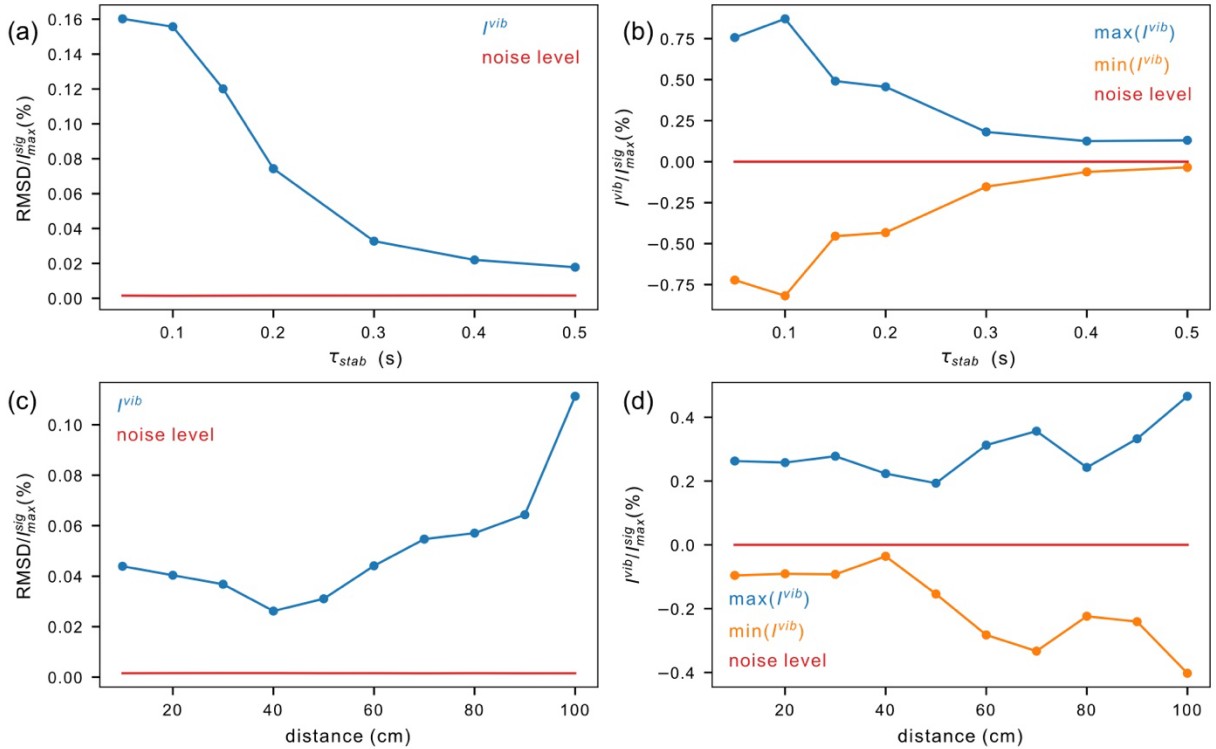

**Figure 11. Amplitude of vibration artifacts in $^1$H NMR spectra of 99.8% D$_2$O (residual water peak 0.2% H$_2$O). Vibrations were**
**quantified in the intervals [-200 Hz, -20 Hz] and [20 Hz, 200 Hz] around the water peak. The level of the noise (red lines) is given as**
**a comparison. (a-b) Decay of vibration artifacts after shuttle transfer from the furthest distance (d$_{max}$ = 1020 mm) at different**
**stabilization times $\tau_{st}$. (a) root mean square deviations of signal amplitudes normalized by the maximum peak intensity; (b)**
**maximum and minimum intensities of vibration artefacts normalized by the maximum peak intensity. (c-d) Extent of vibrations 150**
**ms after shuttle landing as a function of the distance of shuttle displacement d. (c) root mean square deviations of amplitudes**
**normalized by the maximum peak intensity; (d) maximum and minimum intensities of vibration artefacts normalized by the**
**maximum peak intensity.**

### 2.4 Advantages of the design of the fast sample shuttle

During the course of more than 2.5 million shuttling cycles, the system parts show a very low degree of wear and tear. The
sample shuttle container and the rope are disposables that can go through about two million cycles each.

The fast sample shuttle is integrated with the TopSpin software and AU programming (see Figure 5). A single user-friendly
interface can be used and there is no need for external sources to program or control the shuttle units or motors for running
relaxometry experiments. Simple instructions have to be inserted in pulse sequences to initiate the shuttle transfers to low
fields and back. Any type of pulse sequence and all the features of TopSpin can be used. A full relaxation dispersion profile
can be recorded with a single set of instructions. Users create a file listing the target magnetic fields. Either a single list of
relaxation delays can be used for all fields or, if necessary, a separate list of relaxation delays can be used for each magnetic
field.



Sample filling and sample shuttle container preparation are straightforward including the positioning of the plugs. In case of trapped residual air bubbles in the sample volume, the sample shuttle is moved up and down in the shuttle guiding tube to remove the bubbles, making this procedure fast, reliable and simple.

The position of the sample is accurately calibrated when the shuttle sample container reaches the end stopper (12). When, the lower edge of the lower sleeve (14c) touches the soft end stopper, the end stopper exerts an opposite force to that of the working pressure over the container. As a consequence the torque of the motor decreases. This change of torque is a precise measurement of the position of the sample container. For a given value of the torque, the sample is centered at the high-field position and the motor count (position) is stored and used for comparison with the next cycle. This process is automatically

repeated at every cycle to maintain the same the high-field position and compensate for changes of the length of the rope due to temperature variations, humidity, use, etc.. This dynamic calibration makes it possible to run experiments for very long periods of time (days to weeks) with no need to interrupt the experiment for calibration of the position.

Once the shuttle system is installed in the NMR magnet, standard NMR experiments are still possible without deinstallation of the setup. Sealed NMR samples as well as standard 7-inch NMR tube samples can be inserted with the help of a crank

mechanism, which is attached next to the transfer station and uses fitting holders to manipulate the sample. In addition, the crank mechanism is made in such a way that it is not possible to run shuttle experiments while it is in use. Otherwise, the sample shuttle containers can also be used for static NMR experiments.

## 3 Experiments

### 3.1 Comparison with fast field-cycling relaxometry

In order to demonstrate the capability of this fast sample shuttle to measure relaxation rates, we compared magnetic-field dependent relaxation rates measured with the fast sample shuttle and a more conventional approach, fast field-cycling (FFC) relaxometry. In FFC relaxometry, an electromagnet is used, allowing fast switching of the magnetic field, typically within a few ms. However, the magnetic field homogeneity is insufficient to be able to record high-resolution spectra. For the sake of comparison, we measured longitudinal relaxation rates of water protons in solutions of a self-aggregating gadolinium(III) complex and of copper(II) aqua ions at different concentrations using both methods. Longitudinal relaxation rates were

measured with the fast sample shuttle installed on a 700 MHz spectrometer, using a pulse sequence that is equivalent to the "prepolarized" sequence implemented in FFC relaxometry (Figure 12.a). The probe was detuned and small angle RF pulses (10°) were applied in order to avoid problems in quantification of the signal intensities due to radiation damping.

The longitudinal relaxation rates of water protons measured for a 0.25 mM solution of a self-aggregating gadolinium complex in water (Gd-AIE) (Li et al., 2019) are shown in Figure 12.b. The agreement between the data measured with the FFC relaxometer and with the shuttle system is excellent in the overlapping range of frequencies. Measurements with the fast sample



shuttle permit to measure relaxation rates at higher fields than the FFC relaxometer, covering an additional range of static fields from 1 to 16.5 T.

Relaxation rates measured with the shuttle system for the 2.5, 5 and 10 mM copper(II) solutions (Figure 12.c) are also in excellent agreement with the data measured with the FFC relaxometer in the overlapping range of frequencies (only for the 10 mM sample the value at 2 MHz is somewhat smaller when measured with the shuttle). The fit of these data to the Solomon equation is very good (see Figure 12.d), using as fit parameters only a single correlation time (resulting 32 ps) and the copper(II)-water proton distance (resulting 2.7 Å, assuming 6 water molecules coordinated to the copper(II) ion, in fast

exchange with bulk water molecules). On the other hand, faster relaxation is not well characterized with the sample shuttle. With 15 and 20 mM copper(II) solutions, longitudinal relaxation rates at magnetic fields below 10 MHz (or 0.25 T) exceed 25 $s^{-1}$ when measured with the FFC relaxometer. When the sample shuttle is used, the longitudinal relaxation decay rates seem to plateau at ~25 $s^{-1}$. The precision of the measurements is also reduced with errors of about 10% due to the fact that longitudinal polarization is almost at equilibrium at low field when the shortest relaxation delay is used. Overall, these data

demonstrate both that relaxation rates measured with the sample shuttle are in excellent agreement with those measured with conventional relaxometry, when in the range 1 to 20 $s^{-1}$. On the other hand, relaxation rates faster than 20 $s^{-1}$ cannot be measured accurately with the fast sample shuttle apparatus.







**Figure 12. (A) Pulse sequence used for the shuttle measurements. The times τ during which the sample is kept at low fields were set to 16 different values ranging from 0 to 1.75 s. (B) Field dependence of the longitudinal relaxation rates of water protons in a 0.25 mM solution of Gd-AIE at 15 °C and (C) in 2.5, 5, 10, 15, and 20 mM solutions of $Cu(NO_3)_2$ dissolved in a 20 mM $HClO_4$ solution of 90% $H_2O$ and 10% $D_2O$, at 15 °C; the − symbols indicate the rates of the buffer alone. Black symbols indicate data collected with the Stelar FFC relaxometer (0.01-40 MHz) and red symbols indicate data collected at the Bruker 700 MHz spectrometer equipped with the shuttle system (2-700 MHz). (D) Best fit profiles of the relaxation rates of water protons in 2.5, 5, and 10 mM solutions of $Cu^{2+}$ aqua ion.**

### 3.2 Proton high-resolution relaxometry

High-resolution relaxometry is also a sensitive tool to detect the interaction of small molecules with molecular assemblies of larger sizes. Indeed, while the extreme narrowing regime governs most of the nuclear magnetic relaxation dispersion profiles of protons or phosphorus-31 nuclei in a small molecule in solution, a strong dispersion is expected at sub-tesla fields for their relaxation in macromolecules (Pu et al., 2009; Wang et al., 2021). When a small molecule binds transiently to a macromolecule



and exchanges between the free and bound states faster than it relaxes, the nuclear magnetic relaxation dispersion is a population weighted average between those expected for the free small molecule and the complex. Here we use proton relaxometry of a proton in the indole heterocycle of the amino acid tryptophan to probe the transient binding of tryptophan to

human serum albumin. As expected, the NMRD profile of the H2 (or δ1) proton of tryptophan alone in solution is mostly flat (Figure 13.b) thanks to the absence of vicinal scalar couplings (Miesel et al., 2006). On the other hand, the addition of as little as 2.5% equivalent of human serum albumin to the solution leads to a measurable dispersion. Proton relaxometry in small molecules is a demanding application of high-resolution relaxometry, as narrow lines can be visibly distorted by low-frequency vibrations (Figure 13.a). Here, the presence of such small artefacts does not prevent the measurement of relaxation rates of

individual protons.

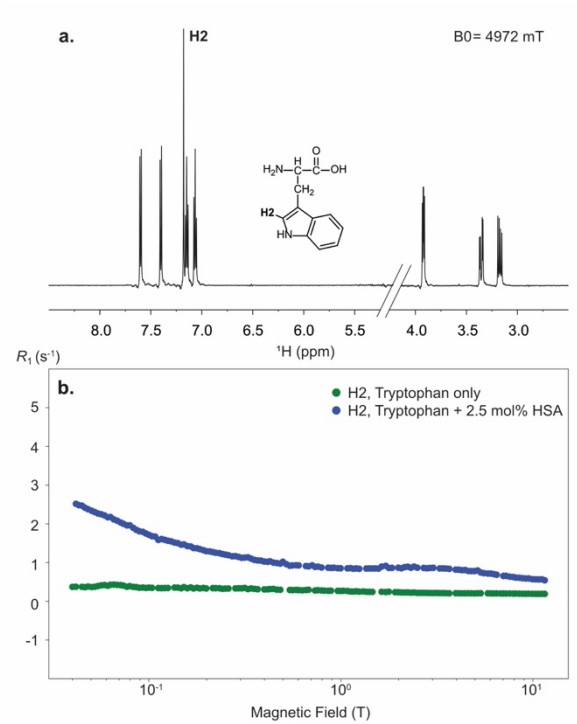

**Figure 13. High-resolution proton relaxometry of tryptophan. (a) Structure and one-dimensional proton spectrum of tryptophan at 14.1 T obtained after a transfer to a magnetic field $B_{low}$ = 4.97 T, a relaxation delay $T_{rel}$ = 3 ms and a stabilization delay at high field**
**$\tau_{st}$ = 150 ms. (b) Nuclear magnetic relaxation dispersion profiles for tryptophan proton H2 in the absence (green) and the presence of 2.5% equivalent of human serum albumin (blue).**

### 3.3 Sample shuttling of an isotopically-labeled protein

The resolution and sensitivity of the fast sample shuttle installed on a conventional high-field NMR spectrometer give access to low-field relaxation of macromolecules. We demonstrated the ability to record high-resolution spectra of a biological

macromolecule on a 200 µM sample of the 42 kDa protein kinase p38γ (Figure 14). The protein was perdeuterated and



selectively labeled on the $\delta_1$ positions of isoleucine residues, one $\delta$ position of leucine residues and one $\gamma$ position of valine residues with $^{13}C^1H^2H_2$ methyl groups. A two-dimensional experiment that corresponds to the shortest delay ($T_{rel}$ = 3 ms) of a longitudinal carbon-13 relaxation at 500 mT is compared to a high-field HSQC spectrum (Figure 14). Both spectra are recorded on the same 600 MHz spectrometer equipped with the fast sample shuttle. The spectral quality of both experiments is identical.

A few peaks are significantly attenuated in the relaxometry spectrum due to faster longitudinal relaxation at low field, during the transfer of the sample container between high and low magnetic fields.

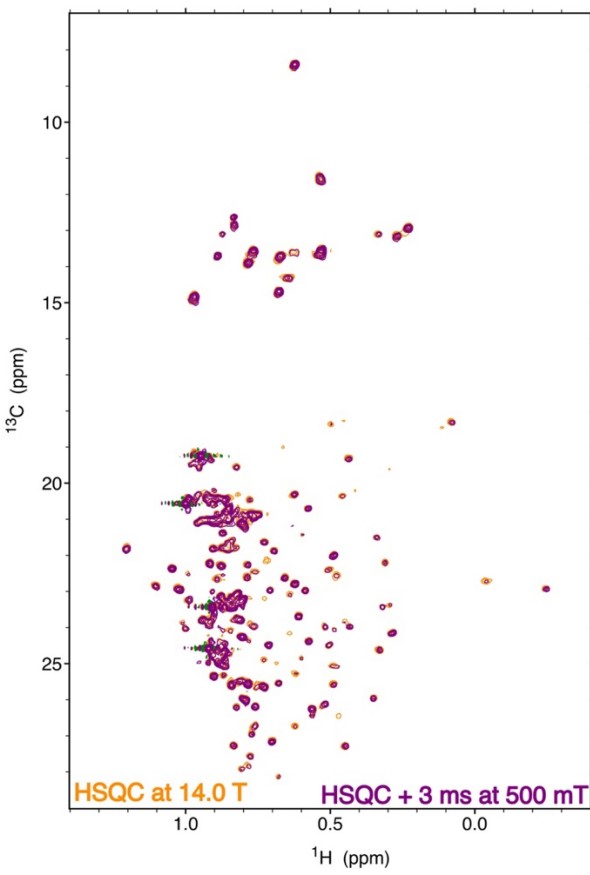

**Figure 14. Comparison of a high-field HSQC spectrum at 14.1 T (orange) and a two-dimensional correlation spectrum obtained at 14.1 T with a transfer to 500 mT (purple). The spectra were recorded on a specifically ILV labeled sample of the protein p38γ with**
**$^{13}C^1H^2H_2$ methyl groups. The HSQC spectrum was recorded with 4 scans and the relaxation experiment with 8 scans. Both experiments were run with 120 complex points in the indirect dimension. The two spectra are displayed with the same contour level.**

## 4 Conclusion

We have introduced a new design of sample shuttle for applications that combine high-field NMR and an evolution at lower magnetic fields. This sample shuttle is installed on a high-field NMR magnet with a conventional high-field probe, providing
state-of-the-art sensitivity and versatile applications. The sample shuttle is a hybrid pneumatic/mechanical design where the

upward force is applied to the shuttle tube by a rope, pulled by a motor and winch wheel and the downward force is applied with a moderate (3 to 4.5 bar) overpressure. With an acceleration up to 113.5 g-units, a speed of 26.7 m.s$^{-1}$ can be reached in *ca.* 27 ms. A transfer delay as short as 61 ms can be achieved to reach a magnetic field of 36.6 mT, 800 mm above the magnetic center of a 14.1 T superconducting magnet. Importantly, the sample shuttle is narrow, which will be adapted for magnetic
manipulation at low field in future set ups. We demonstrate that low-field longitudinal relaxation rates as high as 20 s$^{-1}$ can be measured in solutions of paramagnetic ions in excellent agreement with fast field-cycling relaxometry. We also show nuclear magnetic relaxation dispersion profiles of a single proton in the amino acid tryptophan and obtain high-quality spectra of a 42 kDa protein specifically labeled on methyl groups. The fast sample shuttle offers new opportunities for the development of high-resolution relaxometry and other sample-shuttle applications in NMR.

**Acknowledgements**

We would like to thank for their support, discussions and help Daniel Podadera, Hartmut Glauner, Philipp Aders, Jonas Sommer, at Bruker, Duc-Duy Vu at ENS, and Letizia Fiorucci at CERM.

**Competing interests**

JAVG and AG are employees, FE is a consultant for, JMT is a former employee and AT an employee of a subcontractor of
Bruker Biospin.

**Financial Support**

This work has been supported by the European Commission through the H2020 FET-Open project HIRES-MULTIDYN (grant agreement no. 899683), Marie Sklodowska Curie Action Individual Fellowship HRRinDNAwithSSB (grant agreement no. 101028365) to L. S., and Postdoctoral Fellowship BiophInLLPInt (grant agreement no. 101069121) to U. B. lP., as well as
funding from the "Investissements d'avenir" French National program, project IMF-NMR (ANR-21-ESRE-0041).

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
