# Peer review of "Applications to high-resolution relaxometry"

_Magnetic Resonance, 2024_

## Author Response (AR1)

We thank the reviewers and editor for their positive comments and relevant questions. Please find below our responses to the comments and questions of reviewer 2 and of the Editor. The questions and comments are shown below in italics and our answers start with "A:"

Reviewer 2:

Our answers to reviewer 2 were posted on February 12 by Jorge Villanueva-Garibay. We reproduce them below:

*If I have understood it correctly, the highest point to which the sample can be lifted is "almost [the magnet's] upper closing lid" (line 110) at which point the magnetic field is 36.6 or 46.6 mT for 600 and 700 MHz magnets, respectively. Are lower fields possible? With a suitable modification to the design, could the sample be raised to a point outside the magnet can? If so, would there be space to install a subsidiary magnet that could be used to partially cancel the fringing field of the NMR magnet (as, I believe, has been done in some of the articles mentioned in the Introduction)?*

A: The physical constraints imposed by the magnet geometries have set the current limits. However, the minimum field that can be achieved is not limited to that end and it is indeed possible to reach positions outside the NMR magnet vessel. We are currently working on the technology to add to the current system the capability to get much lower fields and it will be addressed in a forthcoming publication.

*A minor question. Do you have any idea about why the "vibration amplitudes were significantly higher on the FSS system installed on the 600 MHz NMR spectrometer at ENS in Paris" (lines 226-227)?*

A: This is still unfortunately an open question. The 'twin system' at Bruker France has never shown such vibrational sideband artifacts (3 times higher at ENS). Bruker has performed optical, acoustic, magnetic tests in the ENS laboratory, in order to identify possible causes. However, nothing could be concluded from these tests. The main vibrational artifact contribution in the ENS apparatus appears at about 9Hz (very low frequency). It can be (speculation), that this frequency couples with one of the magnet's fundamental frequencies and keeps resonating for a much longer time at higher amplitude.

*Technical corrections Pedantically, the word "rope" suggests something one would use to moor a boat or hang clothes out to dry in the garden. Perhaps "string" or "cord" would more accurately convey an impression of the (presumably) ca. 1 mm diameter length of polyethylene used to lift the sample.*

A: We agree with the reviewer and have replaced the word "rope" with the word "cord" throughout the manuscript. For your information, the cord's diameter is ~ 0.5 mm.

Response to the Editor:

We thank the Editor for his positive comments and address below his questions.

1. *The manuscript states the accessory is compatible with "any high-resolution NMR probehead". If experiments were carried out using a cryoprobe, please state so.*

A: The experiments shown in our manuscript were not performed on a cryoprobe. Most of the experiments were performed on a TXI probe. The experiments shown in Figures 13 and 14 were recorded on a BBFO iProbe. This information will be stated in the revised version of our manuscript at the beginning of the Experiments section. Tests with cryoprobes are work on progress, preliminary results are positive.

2. *It is not clear whether there are limits on the rates at which experiments can be repeated, e.g. due to frictional heating or other limitations. Please include experimental details such as the total measurement time, and how many transients were collected, used for the spectrum of Fig. 14.*

A: As is mentioned in the manuscript, 2.5 million cycles nonstop were achieved (more are still possible and have been achieved). The rate was 2 cycles per second leading to a total test time of about two weeks. We now state the rate of repetition in section 2.4 of our manuscript. Experimentally no temperature effects have been observed in long sessions over few weeks.

The total transients in Fig. 14 are 1024 and the total experiment time was about two hours. The total duration of each experiment was added to the captions of Figures 13 and 14 in the revised version.

3. *The antiphase character of the vibrational sidebands in Fig. 10 suggest oscillations in the tuning of the probe due to transverse sample movement.*

A: The mechanical vibrations artifacts observed in shuttle experiments have the driving motor motion as origin and main contributor. The use of the second counter-rotating motor has substantially diminished but not canceled such artifacts. In addition, the sample landing at the high-field position also contributes to vibrations. We have observed a small transverse sample rotation in a test bench without a probe-head which is due to the applied pressure, but this contribution is negligible.

Early on in this project, we have determined experimentally that vibration artifacts were still present under the motion of motors, even if the sample were not moving and staying at the high field position, with the cord disconnected from the sample

and no pressure applied (see Figure below). These spectra are shown as supporting information.

[Figure]

Figure. 1H NMR spectra of glucosamine showing mechanical vibration artifacts. (blue) FSS setup with one single motor, it defines the vibration's baseline. (red) The cord is disconnected from the winch wheel and the sample is resting at the high-field position, with pressure and motor still active. (green) Pressure is turned off and transfer station removed, leaving only the motor active. (purple) The motor is uncoupled from the supporting structure: no vibrations are transferred anymore.

We have added the following sentences in section 2.3:

The motion of the motor is at the origin of the vibration artefacts. These vibrations remain when the shuttle container is static (with no cord attached) and the motors are moving.

4. *It is not quite clear how the air below the sample is released during the rapid descent of the sample. A comment may be helpful.*

A: The air flow gets out from the high-field turbine the same way as the VT-gas from probe-head.

5. *It appears that friction between the shuttle and the borosilicate glass tube must be minimal to achieve 2 million cycles (ca 4000 km of travel!) without replacement of parts. How is the pneumatic seal achieved without significant friction?*

A: The pressurized gas works also as bearing pressure (similar to rotors in solid-state NMR) keeping the sample container concentric with respect to the guiding tube. This air layer is sufficient to minimize friction as long as the concentricity holds. The transfer station is connected to the pneumatic box to pressurize the system. In addition, it has upper and lower sealing rings that ensure an airtight connection between the transfer station, the motor housing and the upper part of the guidance tube.

We have added the following sentence to our manuscript in the paragraph that accompanies Figure 4:

The flow of the pressurized gas around the sample container ensures the proper orientation of the sample container in the guiding tube, as the bearing gas does for a magic-angle spinning rotor. In addition, the thin layer of gas minimizes contact and hence friction between the sample container and the guiding tube.

6. *Perhaps inclusion of an actual pulse program as Supporting Information, that shows the pp code that is actually used for, say, the HSQC spectrum of Fig. 14 would be helpful to the reader.*

A: We have added the pulse sequence for the measurement of longitudinal carbon-13 relaxation used to obtain the spectrum of Figure 14 in a new supporting information.

Note that the field-cycling commands to run any field-cycling experiment are rather simple. One needs to call an include file with:

**include <Shuttle_fss.incl>**

Then the commands for moving the sample above the probe, followed by a variable delay vd, a transfer back to high-field and a stabilization delay d25 are:

SH_UP
vd
SH_DOWN
d25

*Typo: line 270 "the same the"*

Done.

Fig 2. caption: "at the CERM" while the text immediately above it reads "at CERM".  Probably, the latter is preferred, but in either case, please make the wording consistent.

We have deleted "the".